# Growth Hormone Secretagogues and the Regulation of Calcium Signaling in Muscle

**DOI:** 10.3390/ijms20184361

**Published:** 2019-09-05

**Authors:** Elena Bresciani, Laura Rizzi, Silvia Coco, Laura Molteni, Ramona Meanti, Vittorio Locatelli, Antonio Torsello

**Affiliations:** School of Medicine and Surgery, University of Milano-Bicocca, 20900 Monza, Italy (L.R.) (S.C.) (L.M.) (R.M.) (V.L.) (A.T.)

**Keywords:** calcium (Ca^2+^) homeostasis, GHS (growth hormone secretagogues), cardiac ischemia/reperfusion (I/R) damage, cachexia, skeletal muscle wasting

## Abstract

Growth hormone secretagogues (GHS) are a family of synthetic molecules, first discovered in the late 1970s for their ability to stimulate growth hormone (GH) release. Many effects of GHS are mediated by binding to GHS-R1a, the receptor for the endogenous hormone ghrelin, a 28-amino acid peptide isolated from the stomach. Besides endocrine functions, both ghrelin and GHS are endowed with some relevant extraendocrine properties, including stimulation of food intake, anticonvulsant and anti-inflammatory effects, and protection of muscle tissue in different pathological conditions. In particular, ghrelin and GHS inhibit cardiomyocyte and endothelial cell apoptosis and improve cardiac left ventricular function during ischemia–reperfusion injury. Moreover, in a model of cisplatin-induced cachexia, GHS protect skeletal muscle from mitochondrial damage and improve lean mass recovery. Most of these effects are mediated by GHS ability to preserve intracellular Ca^2+^ homeostasis. In this review, we address the muscle-specific protective effects of GHS mediated by Ca^2+^ regulation, but also highlight recent findings of their therapeutic potential in pathological conditions characterized by skeletal or cardiac muscle impairment.

## 1. Introduction

Calcium (Ca^2+^) is an intracellular messenger that governs a variety of cellular processes such as gene transcription, cell proliferation, programmed cell death, neurotransmission, and muscle contraction and functioning [1]. In skeletal muscle fibers, Ca^2+^ pivotal role is the regulation of the excitation–contraction coupling process, but it is also involved in protein synthesis and degradation, fiber type shifting, activation of Ca^2+^-regulated proteases and transcription factors, and mitochondrial adaptation, plasticity, and respiration [1]. In the myofibers, acetylcholine interaction with its receptor via α-motoneurons generates a depolarizing wave, mediated by voltage-gated sodium channels, which results in L-type Ca^2+^ channel activation (CaV1.1 or dihydropyridine receptor, DHPR). The subsequent entry of Ca^2+^ controls the ryanodine receptors (RyR) in the sarcoplasmic reticulum, allowing a huge release of Ca^2+^ that ultimately favors the cross-bridge between actin and myosin filaments. This process is essential for the sarcomere shortening, the contraction of myofibers, and thus the muscular force generation [1]. 

The maintenance of adequate Ca^2+^ concentrations across extracellular and intracellular compartments of the muscle cell is crucial for sustained contractility. In resting muscle fibers, free Ca^2+^ concentrations are strictly regulated and maintained at low levels (20–50 nM), primarily by the activity of the sarcoplasmic reticulum that accumulates huge amounts of Ca^2+^ ions, which are quickly released in the cytosol upon the arrival of action potentials [1]. The prompt restoration of physiological cytosolic Ca^2+^ concentration is operated, beside sarcoplasmic reticulum, also by the intervention of Ca^2+^ channels, exchangers, pumps, transporters, and binding proteins [1]. Dysregulation of Ca^2+^ signaling/homeostasis represents a common underlying phenomenon in the pathophysiology of many conditions affecting the muscle, such as myopathies and systemic disorders, as well as hypoxia, sepsis, cachexia, sarcopenia, heart failure, and dystrophy [2,3]. 

Growth hormone secretagogues (GHS) are a family of synthetic compounds, characterized by heterogeneous chemical structures, originally discovered for their GH-releasing properties [4]. Later studies demonstrated that GHS mimic the activity of ghrelin, an endogenous 28-amino acid peptide released primarily by the stomach [5,6]. Ghrelin is known as the “hunger hormone” because it stimulates feeding behavior, but it is also involved in the regulation of gastrointestinal motility and energy and glucose homeostasis [5,6]. However, besides the endocrine activities, ghrelin displays a variety of extraendocrine properties shared also by many GHS, targeting peripheral tissues and the central nervous system [7,8,9,10,11,12]. Ghrelin and GHS actions are mainly mediated by binding to the GHS receptor type 1a (GHS-R1a), a G-coupled receptor, and inducing intracellular Ca^2+^ mobilization [13]. Among the peripheral actions, a large body of evidence indicates that ghrelin and GHS improve muscle function in several pathological conditions. Indeed, they show cardioprotective activity both in vitro and in vivo in different experimental models [14,15,16,17,18,19], and humans [20,21]. Interestingly, GHS treatment prevents cisplatin-induced damage in rat skeletal muscles [22,23]. The mechanisms accounting for the beneficial impact of ghrelin and GHS on muscle cells have been only partially elucidated and different hypothesis have been proposed, including the ability to: (i) inhibit apoptosis; (ii) increase nitric oxide (NO) release; (iii) counteract the activity of pro-inflammatory cytokines; (iv) inhibit reactive oxygen species (ROS) generation; and (v) prevent proteasome activity [11,12,22,23,24,25]. In recent years, research has focused on GHS and ghrelin effects on Ca^2+^ homeostasis, speculating that the regulation of this process could play a key role in mediating their specific muscle beneficial activity. This review aims to present the most recent data showing the GHS involvement in Ca^2+^ homeostasis, highlighting their therapeutic potential in different conditions characterized by muscle impairment.

## 2. Growth Hormone Secretagogues (GHS) 

### 2.1. GHS Discovery and State of Art 

The acronym GHS indicates a large family of synthetic compounds with a heterogeneous chemical structure, which includes peptidyl, peptidomimetic, and nonpeptidic moieties. They were originally developed in the late 1970s for their capability to stimulate GH secretion, both in vitro and in vivo, and the great interest toward their endocrine activity ensued from the lack of knowledge of the natural hypothalamic hormone that stimulates GH secretion. Indeed, it is noteworthy that the endogenous growth hormone-releasing hormone (GHRH) was discovered only about 10 years later, in 1982 [26]. Originally, this class of molecules was denominated growth hormone-releasing peptides (GHRP) because of their peptidyl structure, characterized by the presence of tryptophan (Trp) residues [4]; nowadays, these compounds are preferentially designated GHS, in order to include the nonpeptidic compounds, developed in more recent years and endowed with a better oral bioavailability. The first GHRP was synthesized by C.Y. Bowers, starting from the sequence of metenkephalin. Bowers reported that the peptide Tyr-DTrp^2^-Gly-Phe-MetNH2 (DTrp^2^-GHRP), containing dextrorotary (D)-amino acids, weakly stimulated GH release from rat pituitary cells [27]. A few months later, Bowers published the first GHRP with relevant GH-stimulating activity, the hexapeptide His-DTrp-Ala-Trp-DPhe-Lys-NH2, named GHRP-6. GHRP-6 effectively stimulated GH secretion in vitro from pituitary cells, and also in vivo in the rat. Upon GHRH identification, several studies were performed to investigate the relationship between GHRH and GHS. It was concluded that GHS have a dual site of action, at the pituitary and hypothalamic levels, and that their activity was synergic with GHRH, but on different receptors. Results demonstrated that GHS have a mechanism of action involving the activation of phospholipase C, which in turn hydrolyzes the membrane inositol phosphate (IP) into inositol triphosphate (IP_3_) and diacylglycerol (DAG); IP_3_ hence facilitates Ca^2+^ release from intracellular Ca^2+^ stores, causing GH secretion [4]. It soon became clear that the GHS mechanism of action was quite different from that of GHRH, whose binding to its specific seven-transmembrane G protein-coupled receptor (GPCR) activates the cAMP protein kinase A signaling pathway [28]. Further data supporting that GHS and GHRH activate different receptors were provided by this simple demonstration: the first challenge with GHRH blunts the effects of a second stimulation with GHRH, whereas the first challenge with GHS does not affect GHRH effects, and when administered in combination, GHS synergizes with GHRH on GH release [29,30,31,32,33]. Moreover, GHS administration did not activate the physiological negative feedback mechanisms involving somatostatin and insulin-like growth factor-I (IGF-I) release [27]. 

However, although the knowledge of GHS endocrine effects is continuously growing, the comprehension of their exact mechanism of action remains largely unclear. Only twenty years after the discovery of GHRP-6, a specific receptor for the GHS was finally characterized. In 1996, using the radiolabeled peptidomimetic compound MK-0677, a group of researchers at the Merck company found that GHS bind a receptor (GHS-R1a) that is coupled with members of the G_q_/_i_ family, and that activates the phospholipase C, according to early research [13,27]. The GHS-R is synthesized in two isoforms, deriving by an alternative splicing process. The GHS-R1a is a seven-transmembrane G protein-coupled receptor mediating most GHS actions, characterized by a high level of constitutive activity and the ability to form heterodimers with other GPCRs. The GHS receptor 1b (GHS-R1b), instead, is a truncated form of GHS-R1a, encompassing only the first five transmembrane domains. GHS-R1b does not bind GHS and is devoid of any activity on the somatotropic axis, although it is expressed by almost every tissue in the organism [13]. The dichotomy between GHS-R1b wide expression and the failure of promoting GHS biological activity has led to better investigation of its physiological significance, that however still remains largely unclear. Preliminary studies have indicated a possible dominant-negative effect of GHS-R1b, which could rely on its ability to sequestrate the GHS-R1a into intracellular compartments or to stabilize the GHS-R1a in a non-signaling conformation [34]. Nonetheless, new research supports the notion that GHS-R1b could have a more sophisticated activity on GHS-R1a signaling. First, it promotes GHS-R1a trafficking to the plasma membrane with an efficiency that depends on the specific relative GHS-R1b/GHS-R1a expression ratio: increasing the ratio, mobilization of GHS-R1a to the plasma membrane is progressively lost. Second, GHS-R1b impairs GHS-R1a signaling upon oligomerization at the plasma membrane level. Thus, it seems that the relative expression of GHS-R1b not only regulates the efficacy of GHS-R1a signaling, but also assesses the ability of GHS-R1a to form oligomeric complexes with other receptors [34]. 

The endogenous ligand of the GHS-R1a, ghrelin, was unexpectedly isolated from the stomach, where it is predominantly produced by a distinct group of enteroendocrine ghrelinergic cells in the gastric mucosa. Ghrelin is a 28-amino acid peptide that possesses a fatty acid side-chain (preferably C8 or C10) on the serine in position 3. This acylation is essential for binding to GHS-R1a and ghrelin biological functions, including the stimulation of GH secretion and food intake. This rare post-translational modification is achieved by the enzyme ghrelin O-acyl-transferase (GOAT), a member of the membrane-bound O-acyltransferase (MBOAT) family [6,12]. Des-acyl ghrelin (D-AG) or unacylated ghrelin (UAC) is an isoform of ghrelin lacking the acylation and unable to bind the GHS-R1a. Generally considered an inactive product of ghrelin degradation, D-AG has emerged to be an active peptide that, in some cases, shows effects similar to those of ghrelin, but in other instances has different activities [35]. Beginning from the discovery of ghrelin and des-acyl ghrelin, several researches were aimed to explore their physiological role and their involvement in different pathological conditions. However, it became soon evident that their potential clinical use was limited because of the short half-life and the need for parenteral administration. Hence, new agonists and antagonists of the GHS-R1a were developed with better pharmacokinetic profile and oral availability. These ghrelin analogs were intended to be used in specific therapeutic areas such as aging, obesity, cachexia, inflammatory, intestinal and neurodegenerative diseases, muscle, bone and glucose metabolism disorders, cardiac and kidney failure, and to prevent alcohol-seeking behaviors.

In consideration of the pleiotropic peripheral and central effects of ghrelin and ligands of the GHS-R1a, clinical studies are currently ongoing in different pathological fields. Anamorelin and ulimorelin are two agonists of the ghrelin receptor presently under clinical development. Anamorelin is an orally active, high-affinity, selective agonist of the GHS-R1a. Phase 3 clinical studies in patients with advanced non–small cell lung cancer and cachexia reported that anamorelin administration for 12 weeks increased lean body mass (LBM) and body weight, improving some symptoms of anorexia/cachexia [36]. Ulimorelin (TZP-101) is under investigation in diabetic patients with gastroparesis for its capability to stimulate gastric emptying and improve upper gastrointestinal symptoms [37]. On the other side, PF-5190457 is a GHS-R1a inverse agonist currently undergoing clinical development for the treatment of alcohol abuse disorder [38].

### 2.2. GHS Effects on Muscle 

Several severe pathological conditions, from cardiovascular disease to cachexia and aging, recognize muscle damage as a causal or contributing factor [39,40]. Various experimental evidences indicating the capability of GHS to prevent muscle damage or provide a protective outcome in vitro or in vivo have attracted the attention of the scientific community, and a relevant number of researches has been published in recent years on this topic. However, studies investigating the link between GHS protective activity on cardiac and/or skeletal muscle and the regulation of Ca^2+^ homeostasis have focused mainly on two molecules: hexarelin, a well-known peptidyl GHS [22,23,41,42], and JMV2894, a novel peptidomimetic GHS [22,23]. 

#### 2.2.1. GHS Actions on Cardiac Muscle

Cardiac ischemia is the leading cause of mortality in the world. It is primarily caused by a reduction of blood flow in the cardiac coronary arteries [43]. Hence, the primary therapeutic intervention is to restore heart reperfusion. Nevertheless, it is well known that this maneuver can cause additional injury to the cardiac tissue. Ischemia/reperfusion (I/R) damage has been long investigated and entails multiple molecular and biochemical mechanisms, including strong oxidative stress, intracellular Ca^2+^ overload, pH changes, mitochondrial dysfunction, inflammation, and excessive neurohormone release, that predispose cardiomyocytes to apoptosis and necrosis [44]. 

Hexarelin was the first GHS that demonstrated positive inotropic effects in male volunteers [45] and cardioprotective properties against heart injury [46]. in vitro, exposure of neonatal rat cardiomyocytes to hexarelin significantly decreased angiotensin II-induced apoptosis, DNA fragmentation, and increased myocyte viability [47]. The anti-apoptotic activity of hexarelin could also promote survival in H9c2 cardiomyocytes subjected to doxorubicin treatment [48]. Hexarelin cardiotropic activity was also demonstrated in vivo in different experimental models of cardiac damage. Chronic hexarelin administration counteracted the ischemic damage in the heart of Zucker rats subjected to low flow I/R [49], significantly reduced cardiac fibrosis in spontaneously hypertensive rats [50], and showed protective effects against ischemic and post-ischemic ventricular dysfunction in GH-deficient [51] and hypophysectomized rats [15]. In these animal models, several clinical endpoints have been observed, including reductions of left ventricular end-diastolic pressure, coronary perfusion pressure, coronary vasculature reactivity to angiotensin II, and release of creatine kinase and prostacyclin in the heart perfusate, but an enhanced recovery of contractility. Since cardiotropic positive effects were obtained also in GH-deficient and hypophysectomized rats, these actions were specific and presumably independent of somatotropic function, probably caused by hexarelin binding to GHS-R1a and/or CD-36 receptor, a scavenger receptor expressed in cardiomyocytes and microvascular endothelial cells. The involvement of CD-36 receptor in mediating hexarelin action was supported by the lack of hexarelin cardioprotective effects in CD36-null mice and spontaneously hypertensive rats genetically deficient for CD36 [46]. Similar cardiac-positive effects were reported also in humans [52]. Binding of hexarelin to the CD-36 receptor has been proposed as a possible mechanism for its anti-atherosclerotic activities [53]. 

Other studies were performed to explore the capability of GHS to modify intracellular Ca^2+^ homeostasis. Muscle shortening is a function of the force of contraction, and the absence of perfusion could cause post-ischemic areas of dysfunction by causing dissociation between force development and muscle shortening. I/R protocols induce an increase in intracellular [Ca^2+^] transients, possibly due to failure in voltage-gated L type Ca^2+^ channels or alterations of sarcoplasmic reticulum Ca^2+^ content, or both [54]. As proposed by Ma and coworkers, GHS effects in mouse cardiomyocytes exposed to I/R injury are possibly due to several factors. First, GHS treatment administered either before or after the in vitro ischemia protocol significantly ameliorated sarcomere shortening, an index of muscle dynamics, and decreased the amplitude and the rising rate of intracellular [Ca^2+^] transients, indicating the existence of a lusitropic effect [42]. These phenomena are at least in part accounted for in the restoration and maintenance of sarcoplasmic reticulum Ca^2+^ loading ability (SERCA2a), and therefore normal sarcoplasmic reticulum Ca^2+^ loading ability. This hypothesis is also supported by the modulation of phospholamban (PNL) expression [42]. PNL is a small, reversibly phosphorylated, transmembrane protein that is located in the cardiac sarcoplasmic reticulum. It plays a key regulatory role in cardiac contractility: dephosphorylated PNL modulates sarcoplasmic reticulum Ca^2+^ sequestration by inhibiting the SERCA2a. In this form, PLN is an inhibitor of SERCA2a, but, when phosphorylated, PLN dissociates from SERCA2a, activating the Ca^2+^ pump [55]. By increasing the p-PNL/PNL ratio, the GHS favor the activity of SERCA2a, thus suggesting its direct involvement in regulating specific proteins that are part of apparatus of the Ca^2+^ homeostasis setting [42]. It has been suggested that the cardioprotective effects of GHS in vitro could be mediated by the interaction with GHS-R1a, since the pretreatment with two GHS-R1a specific antagonists, [D-Lys3]-GHRP-6 and BIM28163, completely blocked the activity on sarcomere shortening and intracellular [Ca^2+^] transients [42]. From an electrophysiological point of view, the protective actions of GHS are displayed not only by their effects on the Ca^2+^ control system, but also by their capability to preserve electrophysiological properties in I/R condition. It is well known that derangements in Ca^2+^, Na^+^, K^+^ currents and changes in the properties of action potentials are typical of I/R damage [56]. Hexarelin treatment in isolated cardiomyocytes from heart exposed to I/R injury increased action potential amplitude and duration, and preserved many electrophysiological properties. The ability to restore ion currents, in particular reducing inward currents (I_CaL_ and I_Na_) and increasing the transient outward potassium current (I_to_), could also contribute to hexarelin positive inotropic and anti-arrhythmic effects [18]. 

#### 2.2.2. GHS Actions on Skeletal Muscle

Skeletal muscles represent a fundamental element for human life, playing a crucial role in locomotion and respiration. Muscle wasting is related to poor quality of life, increased morbidity, and reduced survival rates in chronic disease states that predispose to poor outcomes and death [57]. The major muscle-wasting disorders are sarcopenia and cachexia [57]. Cachexia is often induced by cancer and frequently complicates the management of patients [58,59]. Although several researches have been focused on this topic, mechanisms underlying muscle wasting in cachexia have not been fully elucidated. Cachexia has been shown to be multifactorial and dependent both on the activation of the intracellular protein degradation apparatus, the ubiquitin (Ub)-proteasome and autophagy systems, and reduced protein synthesis [60]. Given the lack of effective treatments, GHS have been proposed as a potential therapeutic option in the management of cachexia because of their orexigenic and metabolic properties [61]. The cisplatin (CDDP)-treated rat is the experimental model frequently used for studying the molecular mechanisms causing cachexia and to establish the therapeutic value of new candidate drugs. CDDP is a cytotoxic agent widely used in antineoplastic chemotherapy; it has been characterized for its ability to induce in the rat a cachexia condition, with loss of bodyweight and, in particular, of lean muscle mass [62]. It has been demonstrated that short-term treatment with JMV2894 or hexarelin, two ligands of GHS-R1a, accelerated bodyweight gain after the initial loss induced by CDDP treatment. Bodyweight gain was not due to an adipogenic activity, but improving muscle mass and food efficiency, two parameters that indicate, respectively, reduced muscle atrophy and increased energy metabolism [22]. In addition, JMV2894 and hexarelin administration induced a down-regulation of muscle RING-finger protein-1 (Murf-1) E3 transcript expression, a key component of the proteasome system [22,23] and an increase of the transcriptional coactivator peroxisome proliferator-activated receptor γ coactivator-1α (PGC-1α), a marker of muscle oxidative phenotype [23]. Muscle histology analysis revealed attenuated muscle damage, attested by the increased cross-sectional area of myofibers that ultimately lead to improved forelimb force [23]. The beneficial effects of GHS are linked also to their ability to maintain Ca^2+^ homeostasis, counteracting the detrimental effect of CDDP on isolated extensor digitorum longus (EDL) muscles. It is well known that CDDP increases the intracellular resting (Ca^2+^), reduces the response to depolarizing solutions as well as to caffeine, a modulator of the RyR and affects the store-operated Ca^2+^ entry (SOCE), which are essential to ensure proper intracellular [Ca^2+^] for muscle function [23]. Moreover, CDDP alters the expression of genes involved in Ca^2+^ homeostasis apparatus, such as the RyR-type 1(RyR1), the DHPR, the stromal interaction molecule (Stim1) and its main target, the Ca^2+^ release-activated Ca^2+^ modulator 1 (Orai1) [23]. JMV2894 and hexarelin effectively inhibited the Ca^2+^ homeostasis disruption induced by CDDP, reducing the resting intracellular (Ca^2+^), recovering the myofibers responsiveness to caffeine, and thus the RyR activity, and preventing the SOCE decrease. In addition, JMV2894 resulted effective in restoring the expressions of several genes (i.e., RyR1, DHPR, Stim1, and Orai1), being in this context more effective than hexarelin, which only slightly increased the expression of these genes [23]. These different effects of two GHS both displaying agonistic activity on GHS-R1a are somehow not surprising, given the difference in their chemical structure. Indeed, it has been pointed out that minimal changes in the molecular structure of GHS can largely modify their activity; for example, a substitution of a single amino acid residue or using the L- or D-form of the same amino acid can cause loss of the orexigenic activity [63]. On the other hand, the binding sites mediating the GH-releasing and orexigenic effects of GHS are apparently very permissive, since very different peptides (from hexa- to tripeptides) have shown identical activity [63]. Some non-peptidyl GHS can affect Ca^2+^ homeostasis, increasing the resting intracellular (Ca^2+^) differently from hexarelin and other peptidyl compounds in skeletal muscle fibers [64]. This activity could be independent of the GHS-R1a activation, as demonstrated by the use of a specific competitive antagonist of the GHS-R1a, the D-Lys-GHRP-6 [65]. JMV2894 is a peptidomimetic compound with 1,2,4-triazole scaffold as mimetic of the cis-configuration of the amide [65,66]. Although to date only one receptor for GHS has been identified [13], there is a high likelihood that other receptor/s could mediate the plethora of GHS actions, justifying also the different ability of these compounds to interfere with the Ca^2+^ regulation machinery. Moreover, the small molecular size of JMV2894 could suggest its interaction with the RyR and it can exert an RyR stabilizer activity, which is a characteristic of a new class of molecules called Rycals [22,67]. Rycals are capable of reducing Ca^2+^ leak by stabilizing the RyR channels and preserving the RyR–calstabin interaction [68,69]. Rycals have recently been shown to improve contractile function in both heart and skeletal muscle [70]. This peculiarity of JMV2894 could account for the positive effects on sarcoplasmic reticulum responsiveness to caffeine observed in JMV2894 treated rats [22]. 

Recent findings support the hypothesis that JMV2894 could prevent also mitochondrial damage [70]. The regulation of mitochondrial dynamics (fission and fusion) is critical for mitochondrial function, morphology, distribution, and turnover in muscle [71]. It is well known that one of the alterations induced by CDDP is mitochondrial fragmentation, usually related to muscle atrophy in different pathological conditions [71,72]. Muscle function is also highly dependent on communication between sarcoplasmic reticulum and mitochondria, which occurs in a bidirectional fashion by virtue also of their close spatial localization [73,74]. The Ca^2+^ released from sarcoplasmic reticulum is uptaken into adjacent mitochondria to stimulate adenosine triphosphate (ATP) production by aerobic metabolism. ATP, in turn, is required during striated muscle contraction to allow both actin–myosin cross-bridge cycling and to provide the energy to Sarco-Endoplasmatic Reticulum Calcium ATPase- (SERCA) of the sarcoplasmic reticulum during contractile relaxation. On the other end, retrograde mitochondrial–sarcoplasmic reticulum signaling plays a critical role in maintaining proper environmental redox potential, by the production of mitochondrial ROS, and in controlling the local Ca^2+^ activity. In this way, the mitochondria inhibit unwanted Ca^2+^ release around the Ca^2+^ release unit (CRU). CRU is composed of the association of transverse tubule (T-tubule) membrane containing DHPRs, flanked on either side by closely apposed RyR1 (or Ca^2+^ release channels), located in the terminal cisternae of the sarcoplasmic reticulum [71]. The mitochondrial juxtaposition to the CRU in skeletal muscle is allowed by mitofusin 2 (Mfn2), a transmembrane GTPase localized in both sarcoplasmic reticulum and mitochondria outer membranes. Mfn2 is a constituent of electron-dense bridges, or “tethers”, connecting mitochondria to the sarcoplasmic reticulum. Tethers enable highly efficient local mitochondrial Ca^2+^ uptake when the Ca^2+^ concentrations build up in close proximity to sites of Ca^2+^ release [75]. CDDP administration increases Mfn2 protein expression in skeletal muscle and this effect is antagonized by GHS administration [70], suggesting that GHS can exert protective actions on mitochondrial dynamics. 

## 3. Conclusions

Several pathological conditions are characterized by loss and/or impairment of muscle and by muscle wasting. Wasting, characterized by unintentional loss of body weight, encompasses both fat and lean mass and is often the result of renal and cardiac failure, cancer-induced cachexia, and sarcopenia. Dysregulation of Ca^2+^ homeostasis is recognized as one of the causative factors for cachexia and sarcopenia, compromising the functionality of the muscle and predisposing to poor quality of life and progression of the pathology. 

Different therapeutic approaches focusing either on increasing food intake, reversing catabolism and increasing the anabolic drive of cancer patients have been proposed [76]. The few drugs approved for the treatment cachexia are progestins (medroxyprogesterone acetate and megestrol acetate). Progestins improve anorexia and increase body weight, without causing severe side effects. However, different clinical studies have demonstrated that they are not able to increase lean body mass and to improve the patient’s quality of life [77].

The lack of pharmacological effective options in spite of the clinical relevance of these conditions has encouraged the investigation of known and novel GHS to identify new potential therapeutic tools. The use of in vitro cardiomyocytes and the CDDP-induced cachexia model have been useful to study the muscle-specific effects of GHS and to look into their mechanisms of action. Several evidences support the notion that GHS can prevent both cardiac and skeletal muscle damage. The myotropic effects of GHS could be in part due to their ability to prevent Ca^2+^ homeostasis alterations, both in terms of Ca^2+^ movements and expression of specific Ca^2+^-related proteins. On the other hand, the GHS exert protective actions also on the mitochondrial function, indicating that targeting mitochondrial dysfunction may represent an intriguing area of research to develop new pharmacological agents to counteract or limit muscle damage. The increase of mitochondrial biogenesis and the cellular antioxidant defense due to GHS treatment could represent a successful strategy in preventing muscle cachexia. To date, anamorelin, a novel ghrelin receptor agonist, is under clinical evaluation for cachexia treatment because of its ability to improve lean body mass, bodyweight, and in counteracting the anorexia–cachexia symptoms in non–small cell lung carcinoma patients. However, anamorelin administration improved muscle mass but not handgrip strength, a measure of muscle function. Further studies are needed to deepen into aspects related to calcium homeostasis regulation and mitochondrial functioning, to identify new compounds that similarly to hexarelin and JMV2894 could contribute to safeguard in vivo muscle function. Great attention must be directed also to the chemical structure of GHS, as it represents a key feature in selecting the candidate moieties targeted to counteract muscle impairment. 

The Table 1 resumes some interesting GHS and their human physiological effects.

## Figures and Tables

**Table 1 ijms-20-04361-t001:** GHS and human physiological effects.

GHS	Human Physiological Effects
Macrilen (JMV1843) (Macimorelin)	FDA approved for diagnostic test for adult GH deficiency (AGHD).
Anamorelin	Phase 3 clinical trial in on small cell lung cancer: increased lean body mass and bodyweight; improved some symptoms of anorexia/cachexia.
Ulimorelin (TZP-101)	Phase 3 clinical trial in diabetic patients: accelerated gastric emptying and improvd upper gastrointestinal symptoms.
PF-5190457	Phase 1b clinical trial in heavy drinkers: reduced alcohol-primed craving behavior.
MK-0677 – IBUTAMOREN MESYLATE (L-163,191)	Catabolic states: slowed nitrogen wasting and increased fat free mass and bodyweight. GH-deficient adults and hemodialysis patients: increased GH, IGF-1 and prolactin levels throughout the night and ACTH and cortisol levels during the first half of the night.
ARD-07 (EP01572)	Healthy male volunteers: increased GH and IGF-1 release.
Hexarelin	Healthy male volunteers and GH-deficient subject: increased GH and IGF-1 secretion; stimulated food intake. Patients with cardiac dysfunction: cardioprotective effects.
GHRP-2	GH-deficient subject: increased GH and IGF-1 secretion. Healthy male volunteers and obese subjects: stimulated food intake. Anorexia nervosa patients: improved bodyweight and hypoglycemia.
EP80317	Preclinical studies: anti-atherosclerotic activity; anticonvulsivant activity in rat models of status epilepticus; in vitro ACE-inhibiting activity

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
