# Peer review of "Growth Hormone Secretagogues and the Regulation of Calcium Signaling in Muscle"

_ijms, 2019, doi:10.3390/ijms20184361_

Round 1
Reviewer 1 Report
This manuscript is a interesting review about the effects of GHS on muscle impairment.
This manuscript highlight the therapeutic potential of several synthetic compounds derived from Growth Hormone Releasing Peptides (GHRP) ameliorating muscle impairment. The authors pointed out the relevance to link the chemical structure of these derivatives to select the best candidate to apply to muscle diseases".
Author Response
Thank you to reviewer 1 for the revision.
Reviewer 2 Report
This is a short review on the current knowledge of GHS on muscle function and its impact on the Ca handling machinery.
Major
It would help a lot to have a list or table with all currently known GHS. preferentially with their physiological action, and target tissue, etc. to provide a better overview.
On many instances, this reviewer found the manuscript hard to read and would suggest to have an English native speaker to work on a corrected manuscript. Many errors in spelling, word order, etc.
Minor
This reviewer found the use of abbrev, etc quite sloppy throughout the manuscrpt. E.g. wrong spelling, different abbrev for the same entity, …
E.g. cis-platin is commonly abbrev with DDP or CDDP and not CCDP
Phospholamban should be abbrev with either PLB or PLN, but not both. Same for the L-type Calcium channel (either Cav1.1 /Cav1.2 or DHPR; in the latter case the full name should also be given somewhere).
Line 66, which reference belongs where?
Line 76, some italian spelling
Line 84, please explain DTrp
Line 99, full name for IGF-1 is missing
Lind 134, from „showing effects similar…“ on, not clear
Line 204, prevented?
Line 207, should be PLB
Line 254, full name of dhpr is not given
Line 255, „ist main target.“ , not clear
Line 274, is it only the small size that makes it prone to target the RyR?!
Line 307, „cachexia and sarcopenia are recognized….“, this sentence is entirely unclear
Line 309, „ the lack of …“, there is cachexia treatment with progestins
There seem to be formatting issues in the references
Author Response
Replay to Review 2
Major
It would help a lot to have a list or table with all currently known GHS preferentially with their physiological action, and target tissue, etc. to provide a better overview.
A table with some interesting GHS and their physiological effects has been introduced at the end of the paper.
On many instances, this reviewer found the manuscript hard to read and would suggest to have an English native speaker to work on a corrected manuscript. Many errors in spelling, word order, etc .
Thank you for your suggestion. The manuscript has been revised as concern also the english language.
Minor
This reviewer found the use of abbrev, etc quite sloppy throughout the manuscrpt. E.g. wrong spelling, different abbrev for the same entity, ... E.g. cis-platin is commonly abbrev with DDP or CDDP and not CCDP
Phospholamban should be abbrev with either PLB or PLN, but not both. Same for the L-type Calcium channel (either Cav1.1 /Cav1.2 or DHPR; in the latter case the full name should also be given somewhere).
We hypologize for the sloppy mistakes and, as suggested correctly by the reviewer, we revised all the manuscript that now have been armonized in terms of abbrevations and acronyms.
Line 66, which reference belongs where?
The references have been corrected
Line 76, some italian spelling.
The sentence has been amended.
Line 84, please explain DTrp
D-Trp has been specified. (now lines 84-85)
Line 99, full name for IGF-1 is missing
The full name of IGF-1 has been introduced. (now line 102)
Lind 134, from „showing effects similar...“ on, not clear
We have better specified the meaning of the sentence. (now lines 135-136)
Line 204, prevented?
The term “prevented” has beeen replaced. (now line 200)
Line 207, should be PLB
All the acronyms PLB have been amended in the text, as above specified.
Line 254, full name of dhpr is not given
The acronym DHPR has been introduced first in the line 34 of the amended manuscript.
Line 255, „ist main target.“ , not clear
The text has been amended. (now line 253)
Line 274, is it only the small size that makes it prone to target the RyR?!
Both hexarelin and JMV-2894 belong to the GHS family, but their chemical structures are very different. Chemical structure could be responsible for the different efficacy in caffeine responsiveness. Previous experiments with the GHS-R1a antagonists have ruled out the involvement of ghrelin receptor in mediating the effects of GHS on muscle. It was then suggested that the GHS could interact also with other receptors. Conte and co-workers postulated that the small molecular size of JMV2894 could confer a RyR stabilizer-like activity to this molecule. Rycals are generally small molecules. This property is not shared by hexarelin, due to its larger molecular size. However, this is only a suggested hypothesis, and further experiments are needed to support it.
Line 307, „cachexia and sarcopenia are recognized....“, this sentence is entirely unclear
As suggested, the sentence has been amended. (now lines 307-308)
Line 309, „ the lack of ...“, there is cachexia treatment with progestins
As indicated by the reviewer, the progestins mention has been added. (now lines 311-315)
There seem to be formatting issues in the references.
Some issues in References have been formatted.
